# PathVQ: Reforming Computational Pathology Foundation Model for Whole Slide Image Analysis via Vector Quantization

**Honglin Li**[1,2*] **Zhongyi Shui**[1,2*] **Yunlong Zhang**[1,2]
**Chenglu Zhu**[2†] **Lin Yang**[2,3,4†]
[1] College of Computer Science and Technology, Zhejiang University
[2] School of Engineering, Westlake University
[3] The Institute of Advanced Technology, Westlake Institute for Advanced Study
[4] Center for Interdisciplinary Research and Innovation, MuyuanLaboratory
{lihonglin,zhuchenglu,yanglin}@westlake.edu.cn

## Abstract

Pathology whole slide image (WSI) analysis is vital for disease diagnosis and understanding. While foundation models (FMs) have driven recent advances, their scalability in pathology remains a key challenge. In particular, vision-language (VL) pathology FMs align visual features with language annotation for downstream tasks, but they rely heavily on large-scale image-text paired data, which is scarce thus limiting generalization. On the other hand, vision-only pathology FMs can leverage abundant unlabeled data via self-supervised learning (SSL). However, current approaches often use the [CLS] token from tile-level ViTs as slide-level input for efficiency (a tile with 224×224 pixels composed of 196 patches with 16×16 pixels). This SSL pretrained [CLS] token lacks alignment with downstream objectives, limiting effectiveness. We find that spatial patch tokens retain a wealth of informative features beneficial for downstream tasks, but utilizing all of them incurs up to 200× higher computation and storage costs compared [CLS] token only (e.g., 196 tokens per $ViT_{224}$). This highlights a fundamental trade-off between efficiency and representational richness to build scalable pathology FMs. To address this, we propose a feature distillation framework via vector-quantization (VQ) that compresses patch tokens into discrete indices and reconstructs them via a decoder, achieving 64× compression ($1024 \rightarrow 16$ dimensions) while preserving fidelity. We further introduce a multi-scale VQ (MSVQ) strategy, enhancing both reconstruction and providing SSL supervision for slide-level pretraining. Built upon MSVQ features and supervision signals, we design a progressive convolutional module and a slide-level SSL objective to learn spatially rich representations for downstream WSI tasks. Extensive experiments across multiple datasets demonstrate that our approach achieves state-of-the-art performance, offering a scalable and effective solution for high-performing pathology FMs in WSI analysis.

## 1 Introduction

Cancer remains one of the most challenging diseases to diagnose and prognosticate, with pathology playing a pivotal role in understanding its complexities [28]. Traditional histopathological analysis relies heavily on manual examination of tissue samples by pathologists, a process that is not only time-

---

[*]Equal Contribution

[†]Corresponding Author

39th Conference on Neural Information Processing Systems (NeurIPS 2025).

consuming but also prone to inter-observer variability [24]. In recent years, computational pathology has emerged as a transformative method, leveraging whole-slide images (WSIs) to enable automated and quantitative analysis of tissue samples [49, 79, 48]. WSIs, which are high-resolution digital scans of entire tissue slides, provide a wealth of information that can be harnessed for cancer diagnosis, prognosis, and treatment planning. However, the ultra-high resolution of WSIs, often exceeding billions of pixels, presents significant challenges for effective computational modeling [61, 41].

Recent advances in foundation models (FMs)[3, 6, 51, 70] have shown strong potential in computational pathology. Studies have demonstrated the effectiveness of self-supervised learning (SSL)[75, 10, 20, 52] and vision-language (VL) pretraining [17, 48, 57] in extracting semantic features on pathology images. The FMs typically process WSIs by dividing them into smaller tiles (e.g., 224×224 pixels as a tile), extracting features from each tile, and aggregating these features to make slide-level predictions. VL-FMs [57, 48] excel in downstream tasks (e.g. zero-/few-shot ROI classification [65]), but their scalability is limited by the scarcity of large-scale image-text pairs [29, 65, 30]. In contrast, vision-only FMs trained on unlabeled data via SSL are more scalable. However, most methods adopt the task-agnostic [CLS] token from pretrained ViTs as a global representation of each tile [48, 20] and fed as WSI input [31, 61, 10, 33, 40, 62, 43, 67]. This approach overlooks critical spatial information captured by other spatial tokens, which are particularly essential for modeling nuanced pathological variations in gigapixel WSIs.

Notably, some studies have attempted to address this issue by scaling up to larger models (UNI-2 [10] using ViT-giant) or combining [average pooling] features with the [CLS] token (Virchow-2 [90]). Disappointingly, these approaches yield only marginal improvements. Consequently, we argue that the scalability and performance of FMs is fundamentally constrained by the trade-off between efficiency (using [CLS] only) and representational richness (using all patch tokens):

Leveraging all spatial patch tokens benefit WSI analysis but incurs nearly 200× higher storage and training costs as shown in Figure 1 (e.g., 196 tokens in $ViT_{224}$). To address this, we introduce feature distillation via vector quantization (VQ) [71, 53] on patch features, which efficiently compresses spatial patch tokens using discrete indices and a decoder. Our method reduces token dimensionality from 1024 to 16, achieving a 64× compression rate while preserving reconstruction fidelity. This compression process retains original spatial and contextual information, ensuring that critical features are preserved for downstream tasks.

Furthermore, we employ a multi-scale VQ (MSVQ) strategy, which unifies patch-level and tile-level feature VQ. Intuitively, tile-level feature like [CLS] token can be seen adaptive combination all patch features, thus they share the same feature space and

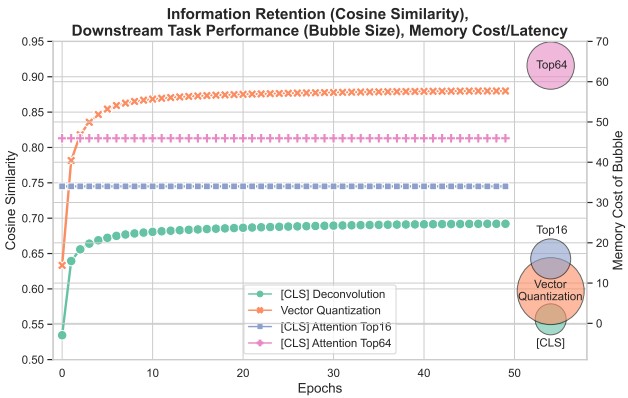

Figure 1: Evaluation on information loss via reconstruction training. Directly using the [CLS] token results in significant information loss, making it difficult to reconstruct all patch tokens, potentially discarding critical details for downstream tasks. In contrast, vector quantization retains more original information and show stronger result on downstream task 'BRACS'. The TopK patch tokens via CLS tokens attention selection are included for comparison.

can be learned into a single VQ model. The MSVQ not only enhances VQ reconstruction performance but also serves as a SSL supervision target for a seamless slide-level pretraining objective (working as a tokenizer thus can be pretrained like BERT [15, 25, 53]). By integrating slide SSL into our framework, we enable the model to learn rich, discriminative representations from unlabeled WSIs, addressing the challenge of limited WSI samples in computational pathology downstream tasks. Built upon the quantized features of patches and supervision targets of tiles via MSVQ, we develop a progressive convolutional module and slide-level SSL to extract representations with rich spatial information for downstream WSI tasks, leading to more accurate and interpretable predictions for

tasks like cancer diagnosis and prognosis. The contributions of our work can be summarized as follows:

1) Efficient Token Compression with VQ Distillation: We propose a novel VQ-based framework that compresses patch-level spatial tokens by $64\times$ while retaining critical spatial and contextual information, enabling scalable and efficient WSI analysis.

2) SSL Supervision via offline tokenizer: Our improved MSVQ strategy not only enhances feature reconstruction but also serves as an SSL supervision target for slide-level mask prediction, providing a new direction for pretraining WSI models.

3) Rigorous Validation: Extensive evaluations on multiple datasets demonstrate the effectiveness of our approach, achieving state-of-the-art performance in WSI analysis tasks, with practical implications for clinical applications.

By addressing the computational challenges of WSI analysis while preserving critical spatial information, our framework offers a new perspective on the development of computational pathology foundation models, paving the way for more accurate and scalable cancer diagnostics.

## 2 Method

### 2.1 Preliminary

For WSI modeling, a WSI $\mathbf{X}$ is first divided into $N$ tiles: $\mathbf{X} = [\mathbf{x}_1, \mathbf{x}_2, \ldots, \mathbf{x}_N]$, which are then processed by the FM. The pretrained FM ViT converts tile image $\mathbf{x}$ into $n$ patches $\mathbf{x} = [\mathbf{p}_1, \mathbf{p}_2, \ldots, \mathbf{p}_n]$, where the most commonly used patch size is $16 \times 16$. The ViT outputs all patch representations within a tile: $[\mathbf{s}; \mathbf{h}_1, \mathbf{h}_2, \ldots, \mathbf{h}_n]$, where $\mathbf{s}$ serves as a summary [CLS] of the spatial tokens of all patches ($\mathbf{ST} = [\mathbf{h}_1, \mathbf{h}_2, \ldots, \mathbf{h}_n]$). Most existing approaches [31, 84, 10] rely on the [CLS] token from each tile to form WSI input embeddings $\mathbf{S} = [\mathbf{s}_1, \mathbf{s}_2, \ldots, \mathbf{s}_N] \in \mathbb{R}^{N \times D}$. These embeddings are subsequently aggregated for slide-level prediction: $\hat{Y} = g(\mathbf{S}; \theta)$, where $g(\theta)$ can be an attention [31] mechanism or a Transformer.

In contrast, this paper explores using all spatial tokens $\mathbf{H} = [\mathbf{ST}_1, \mathbf{ST}_2, \ldots, \mathbf{ST}_N] \in \mathbb{R}^{N \times n \times D}$ for slide-level prediction. Here, $N$ (the number of tiles) can easily exceed 5k, $n = 196$ (the number of patches per tile), and the feature dimension $D = 1024$ (for UNI [10]). So, directly leveraging these high-dimensional data (about 1 million patch tokens) is computationally prohibitive for WSI training.

### 2.2 Vector Quantization Learning

To mitigate the computational burden while incorporating all patches' ST representations, we introduce vector-quantization (VQ) learning on the pretrained FM's patch ST, as illustrated in Figure 3b. This framework consists of an encoder, a quantizer, and a decoder. Additionally, we extend VQ to support both patch and tile representations via a multi-scale VQ strategy.

#### 2.2.1 VQ for Patches

The spatial tokens ($\mathbf{ST}$) are mapped into discrete codes through vector quantization (VQ). Specifically, the tile-level representation $\mathbf{ST} = [\mathbf{h}_1, \mathbf{h}_2, \ldots, \mathbf{h}_n]$ is first passed through an MLP encoder to reduce its dimensionality from $D$ to $d$:

$$[\mathbf{e}_1, \mathbf{e}_2, \ldots, \mathbf{e}_n] = \mathrm{Enc}([\mathbf{h}_1, \mathbf{h}_2, \ldots, \mathbf{h}_n]), \tag{1}$$

where the resulting low-dimensional representations $[\mathbf{e}_1, \mathbf{e}_2, \ldots, \mathbf{e}_n]$ are subsequently tokenized into discrete indices $\mathbf{ST}_{\mathrm{tok}} = [z_1, z_2, \ldots, z_n]$. The codebook $\mathbf{V} = [\mathbf{v}_1, \mathbf{v}_2, \cdots, \mathbf{v}_C] \in \mathbb{R}^{C \times d}$ consists of $C$ learnable embeddings. Each patch-level representation $\mathbf{e}_i$ is assigned to its nearest neighbor in the codebook via:

$$z_i = \arg\min_j \|\ell_2(\mathbf{e}_i) - \ell_2(\mathbf{v}_j)\|_2, \tag{2}$$

where $j \in \{1, 2, \ldots, C\}$, and $\ell_2$ denotes $L_2$ normalization used for distance computation, ensuring that each patch token is matched to the most similar codebook vector.

After quantization, the selected embeddings $\mathbf{E}_{z_1}, \mathbf{E}_{z_2}, \ldots, \mathbf{E}_{z_n}$ are passed to a multi-layer Transformer decoder to reconstruct the original spatial token representation. During training, the decoder output $\mathbf{o}_i$ is aligned with the target $\mathbf{h}_i$ by maximizing their cosine similarity.

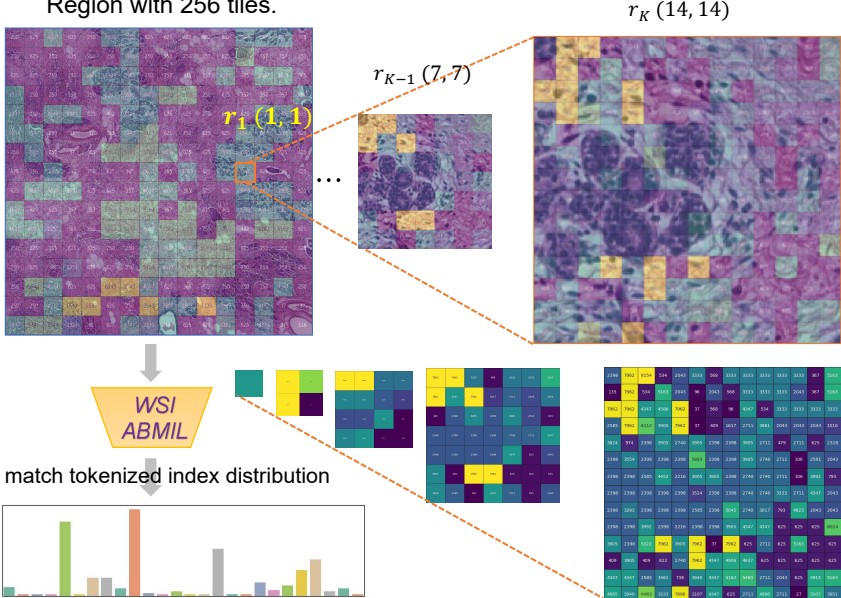

Figure 2: Multi-scale Vector Quantization (MSVQ) visualization. Based on MSVQ, the tile- and patch-level can be quantified simultaneously for slide-level pretraining and feature compression, respectively. The region data can be used to pretrain ABMIL via token index frequency matching.

Since the quantization operation in Equation 2 is non-differentiable, we adopt the straight-through gradient estimator [71], which copies gradients from the decoder input to the MLP encoder output for backpropagation.

The overall training objective of VQ is defined as:

$$\max \sum_{x \in M} \sum_{i=1}^{n} \cos(\mathbf{o}_i, \mathbf{h}_i) - \|\ell_2(\mathbf{e}_i) - \ell_2(\mathbf{v}_i)\|_2^2, \tag{3}$$

where $M$ denotes the dataset of image tiles. For simplicity, we omit the straight-through gradient path and stop-gradient notation [71]. During optimization, the MLP encoder, codebook embeddings, and Transformer decoder are jointly trained to reconstruct the original spatial token representations.

### 2.2.2 Multi-Scale Vector Quantization

To simultaneously compress patch-level spatial token $\mathbf{ST}$ representations and generate an offline tokenizer for WSI self-supervised learning, we propose a **Multi-Scale Vector Quantization (MSVQ)** module. MSVQ encodes the FM tile $\mathbf{ST}$ features into $K$ multi-scale discrete token maps $\mathbf{R} = (\mathbf{r}_1, \mathbf{r}_2, \ldots, \mathbf{r}_K)$.

MSVQ builds upon the VQ architecture described in Section 2.2.1, with the key addition of a multi-scale quantization module. The encoding process is designed with residual paradigm [68, 38], as detailed in Algorithm 1.

Intuitively, when $\mathbf{R} = (\mathbf{r}_1)$ as scale ratio, MSVQ reduces to a standard VQ applied to the average-pooled patch token representation—akin to the tile-level [CLS] token. On the other hand, when $\mathbf{R} = (\mathbf{r}_K)$, MSVQ behaves identically to the patch-level VQ introduced in Section 2.2.1. The general form $\mathbf{R} = (\mathbf{r}_1, \ldots, \mathbf{r}_K)$ enables vector quantization at multiple-scale semantic levels, including tile, patch, and intermediate resolutions. Please refer to Figure 2 for a visual illustration.

A shared codebook $\mathbf{Z}$ is employed across all scales, ensuring that tokens from each $\mathbf{r}_k$ are drawn from a unified vocabulary $[V_C]$. The decoding process mirrors the encoding pipeline in reverse order.

---
**Algorithm 1:** Multi-Scale VQ Encoding
---
**1 Inputs:** FM's spatial token feature $\mathbf{ST} = [\mathbf{h}_1, \mathbf{h}_2, \ldots, \mathbf{h}_n]$ # $\mathbf{h}_i$ is the ViT outputs tokens despite of CLS;
   **Hyperparameters:** number of scales $K$, resolutions $(H_k, W_k)_{k=1}^K$;
**2** $\mathbf{f} \leftarrow \text{Enc}(\mathbf{ST})$ is encoded feature, $\mathbf{R} \leftarrow []$ represents the residual list;
**3 for** $k = 1, \ldots, K$ **do**
**4**   $\mathbf{r}_k \leftarrow \mathcal{Q}(\text{interpolate}(\mathbf{f}, H_k, W_k))$, # get residual of resolution level k;
**5**   $\mathbf{R} \leftarrow \text{queue\_push}(\mathbf{R}, \mathbf{r}_k)$;
**6**   $\mathbf{z}_k \leftarrow \text{lookup}(\mathbf{Z}, \mathbf{r}_k)$;
**7**   $\mathbf{z}_k \leftarrow \text{interpolate}(\mathbf{z}_k, H_k, W_k)$;
**8**   $\mathbf{f} \leftarrow \mathbf{f} - \mathbf{z}_k$;
**9 Return:** multi-scale tokens $\mathbf{R}$ and codebook indices $\mathbf{Z} = [\mathbf{z}_1, \ldots, \mathbf{z}_K]$;
---

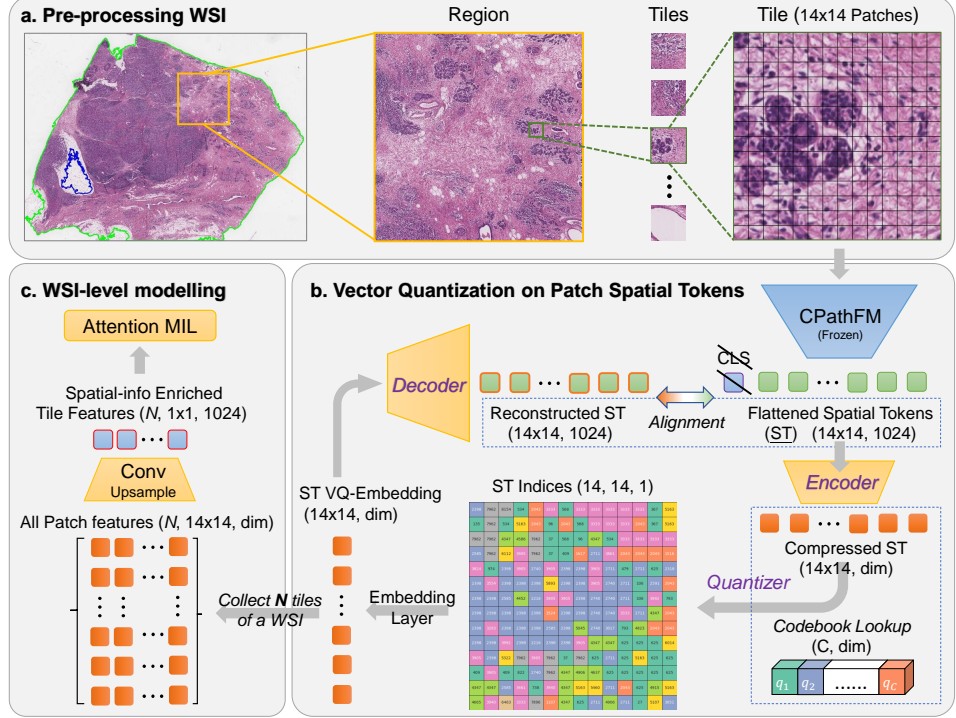

Figure 3: Overview of the proposed framework. (a) The pipeline for compressing spatial patch tokens using vector quantization (VQ) and multi-scale VQ (MSVQ). (b) Slide-level self-supervised learning (SSL) using MSVQ-generated tokenizers. (c) Downstream WSI task fine-tuning with compressed patch features.

## 2.3 Slide-Level Self-Supervised Learning

Leveraging the offline tokenizer generated by MSVQ for all WSI tiles, we design a self-supervised learning (SSL) pretraining framework tailored for WSI-MIL analysis. This framework is compatible with both mainstream MIL architectures, including attention-based MIL (ABMIL) and Transformer-based models.

### 2.3.1 ABMIL-Based Self-Supervised Learning

In supervised ABMIL training (e.g., WSI classification), adaptive pooling or max-pooling is typically employed to aggregate tile-level features for prediction at the WSI level. Inspired by this paradigm, we formulate a simple yet effective SSL objective for ABMIL, grounded in the level-1 quantized indices from MSVQ (as shown in Figure 2).

Given a large-region crop from a WSI (e.g., a region of size $14336 \times 14336$, corresponding 4096 tiles each sized $224 \times 224$), the SSL objective for each region $\mathbf{x}$ is defined as:

$$\mathcal{L}(\boldsymbol{\theta}) = -\sum_{c=1}^{C} q_c(\mathbf{x}) \log p_{\boldsymbol{\theta}}(\mathbf{x})_c, \tag{4}$$

where the soft target distribution $q_c(\mathbf{x})$ is computed based on the frequency of MSVQ token indices within the region, normalized over all $C$ codebook categories. Specifically, $q_c(\mathbf{x})$ represents the proportion of tiles in region $\mathbf{x}$ assigned to token class $c$. The predicted probability $p_{\boldsymbol{\theta}}(\mathbf{x})_c$ for class $c$ is obtained by passing the region through an ABMIL model followed by a classifier head with softmax activation:

$$p_{\boldsymbol{\theta}}(\mathbf{x})_c = \text{softmax} \left( \text{classifier} \left[ \text{AttnPool}(\mathbf{x}) \right] \right)_c. \tag{5}$$

### 2.3.2 WSI Transformer-Based Self-Supervised Learning

We adopt a masked image modeling (MIM) strategy inspired by MAE [25] and BEiT [53], but with a key difference: instead of raw image patches, we operate on pre-extracted feature representations as input. Given an input region composed of $k$ tiles, represented as $\mathbf{x} = \{\mathbf{t}_1, \mathbf{t}_2, \ldots, \mathbf{t}_k\}$, we randomly mask a subset of tiles indexed by $\mathcal{M}$. The masked positions are replaced with a shared learnable embedding $\mathbf{e}_{[M]}$, and Rotary Positional Embedding (RoPE) [64] is applied to retain spatial coherence. The corrupted input becomes:

$$\mathbf{x}_{\text{corrupt}} = \{\mathbf{t}_1, \cancel{\mathbf{t}_2}, \ldots, \cancel{\mathbf{t}_i}, \mathbf{t}_{i+1}, \ldots, \mathbf{t}_k\}. \tag{6}$$

For each masked tile, a softmax classifier is trained to predict the corresponding token index, which is obtained from the level-1 quantized output of the MSVQ tokenizer (see Section 2.2.2). This provides a discrete and consistent supervision signal.

The training objective is formulated as:

$$\mathcal{L}_{\text{mask-modeling}}(\boldsymbol{\theta}) = -\sum_{\mathbf{x} \in \mathcal{D}} \sum_{i \in \mathcal{M}} \log p_{\boldsymbol{\theta}}(z_i \mid \mathbf{x}_i^{\mathcal{M}}), \tag{7}$$

where $z_i$ denotes the MSVQ token index for the $i$-th masked tile, and $\mathcal{D}$ is the dataset of training regions. Compared to the online tokenizer used in iBOT and related frameworks, our MSVQ-based offline tokenizer provides a more stable and reliable supervisory signal for SSL pretraining.

### 2.4 WSI Downstream-Task Fine-Tuning

As illustrated in Figure 3c, the refined WSI input consists of patch feature embeddings with a compressed shape of $(N, 14, 14, \text{dim})$, where $N$ represents the total number of tiles in a WSI, and $(14, 14)$ corresponds to the standard 2D patch arrangement in ViT for each tile. To enhance the feature representation for downstream tasks, we first apply convolutional layers (Convs) with upsampling, increasing the output channel size while reducing to fewer tokens to better capture task-relevant information. The extracted features are then reshaped to match the original CLS token representation of tile-based ViT.

It is notable that the encoding direction of $\mathbf{ST}$ is not so controllable since the the encoded embeddings are in the middle layers (after encoding, before decoding). To keep its feature space as original, we align the output of Convs to original level-1 tile feature during VQ pre-training. This module will be further fine-tuned during slide-level task. Finally, the processed features are fed into downstream MIL models, including both ABMIL and WSI-Transformer (see Section 2.1).

### 2.5 Overall Framework and Implementation

We summarize the overall framework of our method below. The WSI pre-processing follows the approach used in previous work [49]:

- **Patch-Level VQ Learning (Figure 3b)**: This module aims to compress all patch token features from FMs, making them trainable for downstream tasks. Multi-scale VQ learning (Figure 2) further enables slide-level SSL supervision and subsumes patch-level VQ learning.

- **Slide-Level SSL (Figure 2)**: Leveraging the tile-level tokenizer learned via MSVQ, SSL can be effectively applied to both ABMIL and WSI-Transformer models.

- **WSI Downstream-Task Fine-Tuning (Figure 3c)**: Fine-tuning serves two purposes: (a) transforming patch features into a more suitable representation for downstream tasks, and (b) fine-tuning the pretrained slide-level SSL model for improved performance.

# 3    Experiments

In this section, we evaluate the performance of the proposed method and compare it with various baselines. Additionally, we conduct ablation studies to further analyze its effectiveness.

## 3.1    Pretraining Implementation Details

**VQ Pretraining:** We conduct VQ pretraining on 1M randomly cropped $224 \times 224$ tiles extracted from all TCGA [69] diagnostic pathology WSIs. During training, the FM backbone (e.g., UNI with ViT-Large) remains frozen. The codebook has a size of $C = 8192$ with an embedding dimension of 16. For MSVQ, we employ a multi-scale resolution list: $\{1 \times 1, 2 \times 2, 4 \times 4, 7 \times 7, 14 \times 14\}$. The VQ encoder, decoder, and codebook are frozen after pretraining. The model is trained on 4 RTX-3090 GPUs for 50 epochs using a batch size of 128 tile images per GPU. The total training time is approximately 22 hours.

**WSI-SSL Pretraining:** We crop all TCGA diagnostic WSIs into regions of resolution $3584 \times 3584$, yielding a dataset of approximately 250k regions. To facilitate SSL, a pretrained MSVQ model is used to extract the quantized indices of each tile within a region, requiring only about 65MB for storage.

During pretraining, the indices of each region are first re-embedded via a frozen VQ module, resulting in a feature representation of shape $(256, 14 \times 14, 16)$. The convolutional module consists of four conv layers with a stride of 2, progressively increasing the output channels from $128 \rightarrow 256 \rightarrow 512 \rightarrow 1024$. This process transforms the features into spatially enriched embeddings of shape $(256, 1 \times 1, 1024)$. These embeddings are subsequently fed into either an ABMIL model or a 6-layer WSI-Transformer for pretraining.

## 3.2    Downstream Tasks

We primarily focus on WSI classification and survival prediction. For dataset details, please refer to Appendix A.2. For illustration purpose, we also run two experiments on ROI classification to clarify [CLS] token is not all we need.

The data processing and embedding procedure are identical to the region-based approach but are applied at the WSI level. The $(x, y)$ coordinates of tiles are also stored to facilitate positional encoding in the WSI-Transformer.

During fine-tuning, both the convolutional module and the WSI model are trained with a batch size of 1 for 20 epochs. The learning rate is fixed at $1 \times 10^{-4}$, with a weight decay of $1 \times 10^{-4}$, using the AdamW optimizer with default settings.

For ABMIL, both randomly initialized and pretrained models are fine-tuned using the same hyper-parameters and training protocol. For WSI-Transformer, LoRA [27] adaptation (applied to all `nn.Linear` layers) is used with $rank = 16$ during fine-tuning of the pretrained model to mitigate overfitting. For Transformer initialized from scratch, full fine-tuning is employed.

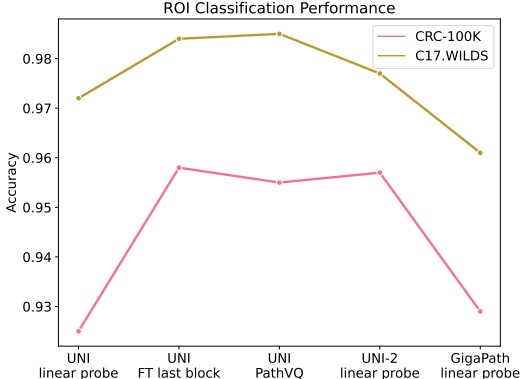

Figure 4: ROI classification. Obviously, further fine-tuning (FT) on the last block of UNI ViT can further improve the downstream results. PathVQ, by compressing and reconstructing the patch spatial token, can achieve comparable improvement. UNI-2, however, does not show consistency improvement compared to FT and PathVQ.

### 3.2.1 Tile/ROI Classification

We evaluate tile/ROI classification performance using dataset CRC-100K [35] (9 categories) and Camelyon-17 WILDS [36] tiles (binary). The result in shown in Figure 4: By only updating the last Transformer block of UNI, the result can be significantly improved. Our PathVQ method is also included and shown comparable improvement to FT. All these results are better than linear probe (freeze backbone and fed [CLS] token feature to classification head). UNI-2, also using linear-probe seems can not scaling up with strong performance on every down-stream tasks.

### 3.2.2 WSI Tumor Classification

Table 1: Slide-Level Tumor Classification based on FM. The results in the first-row are all trained on UNI, while the second-row we include some recent stronger FMs. The cyan rows are our methods including PathVQ and Slide-level Pre-Training (SPT). The orange rows demonstrate how much ($\Delta$) of our PathVQ method and UNI-2 improved over UNI with ABMIL setting. The **bold** and underline denote the best and second-best result, respectively.

| Method | Tumor classification BRACS | | Mutation Prediciton LGG-GBM | |
|---|---|---|---|---|
| | F1 | AUC | F1 | AUC |
| CLAM-SB [49] | $0.640_{\pm0.05}$ | $0.844_{\pm0.03}$ | $0.672_{\pm0.06}$ | $0.842_{\pm0.03}$ |
| DTFD-MIL [84] | $0.655_{\pm0.03}$ | $0.878_{\pm0.02}$ | $0.697_{\pm0.04}$ | $0.857_{\pm0.02}$ |
| TransMIL [61] | $0.592_{\pm0.03}$ | $0.859_{\pm0.02}$ | $0.678_{\pm0.05}$ | $0.847_{\pm0.03}$ |
| ABMIL [31] | $0.692_{\pm0.03}$ | $0.875_{\pm0.02}$ | $0.685_{\pm0.07}$ | $0.852_{\pm0.04}$ |
| +PathVQ | $0.730_{\pm0.02}$ | $0.902_{\pm0.01}$ | $0.723_{\pm0.04}$ | $0.871_{\pm0.04}$ |
| $\Delta$ over UNI + ABMIL | 3.8% ↑ | 2.7% ↑ | 4.8% ↑ | 1.9% ↑ |
| +PathVQ + SPT | $\underline{0.747}_{\pm0.01}$ | $\underline{0.906}_{\pm0.01}$ | $\underline{0.752}_{\pm0.03}$ | $\underline{0.879}_{\pm0.02}$ |
| Roformer | $0.678_{\pm0.03}$ | $0.882_{\pm0.01}$ | $0.675_{\pm0.03}$ | $0.861_{\pm0.02}$ |
| +PathVQ | $0.711_{\pm0.02}$ | $0.892_{\pm0.01}$ | $0.739_{\pm0.04}$ | $0.872_{\pm0.02}$ |
| +PathVQ + SPT | $\mathbf{0.754}_{\pm0.02}$ | $\mathbf{0.910}_{\pm0.01}$ | $\mathbf{0.758}_{\pm0.02}$ | $\mathbf{0.886}_{\pm0.01}$ |
| UNI-2 + ABMIL | $0.698_{\pm0.03}$ | $0.887_{\pm0.02}$ | $0.699_{\pm0.03}$ | $0.859_{\pm0.01}$ |
| $\Delta$ over UNI + ABMIL | 0.6% ↑ | 1.2% ↑ | 1.4% ↑ | 0.7% ↑ |
| GigaPath | $0.677_{\pm0.03}$ | $0.862_{\pm0.03}$ | $0.703_{\pm0.04}$ | $0.864_{\pm0.02}$ |
| TITAN | $0.696_{\pm0.04}$ | $0.891_{\pm0.01}$ | $0.711_{\pm0.03}$ | $0.868_{\pm0.02}$ |

We first evaluate our method on the BRACS [4], a dataset with three categories—negative, benign, and malignant cancer. We then evaluate on TCGA LGG-GBM [69] focus on R132 [2] gene mutation as binary classification. (We notice that popular-used WSI binary tumor classification tasks(e.g. Camelyon [47], TCGA-NSCLC [69]) are nearly solved (AUC>97) given FMs progress. So here we mainly focus on more difficult task, like more categories, and will explore and validate more difficult datasets in near future.)

**Compared Baselines:** Since our method primarily focuses on extracting improved tile-level features for WSI analysis, we compare it against various WSI analysis models with different architectural designs: ABMIL [31], DSMIL [39] (introduces a max-pooling branch alongside the attention mechanism), and DTFD-MIL [84] (employs sub-bags for hierarchical learning). TransMIL [61] (leveraging Nyström self-attention [78] for computational efficiency), Transformer with 2-d RoPE [64, 41, 54].

FMs like GigaPath (a 12-layers WSI-Transformer (efficiently implemented using LongViT [74]), pretrained on large-scale private data via MAE [25], with a [CLS] token as tile feature), TITAN [79, 17] (a 6-layers WSI-Transformer with 2D-ALiBi positional encoding [55, 41], pretrained on large-scale private data using iBOT [88], with Conch-v1.5 as the tile feature extractor ([CLS] token).), and UNI-2 are also included. Certain works that focus on orthogonal aspects, such as overfitting mitigation, hard instance mining, etc. [89, 86, 87, 56, 66, 67, 14, 45, 80], are not included in our primary comparison.

For all the experiments, we report the macro-AUC and macro-F1 scores (over five-runs or five-fold cross validation) because of class imbalance.

**WSI Classification Results Analysis:** The results are reported in Table 1. We can first observe that ABMIL and Roformer show significant improvement when combined with our PathVQ compressor into UNI. The results difference of UNI+PathVQ+ABMIL (about 3% improvement with adding 1M tile data) and UNI2+ABMIL (about 1% improvement with adding large-scale ($>>$1M) of tile data, and $2 \sim 6\times$ model size) demonstrate that the scalability of previous FMs are bottlenecked by the [CLS] token information losses. In addition, the results of our slide-level pretraining (**SPT**) also show consistency improvement compared with random initialization.

### 3.2.3 WSI Survival Prediction

Table 2: **Survival prediction** Results of PathVQ and baselines for measuring patient disease-specific survival. All methods in Prototype and MIL use UNI features [10]. Best performance in **bold**, second best underlined.

| | TCGA | BRCA | CRC | BLCA | UCEC | KIRC |
|---|---|---|---|---|---|---|
| **Prototype** (unsup. cox loss) | H2T [73] | $0.672_{\pm0.07}$ | $0.639_{\pm0.11}$ | $0.566_{\pm0.05}$ | $0.715_{\pm0.09}$ | $0.703_{\pm0.11}$ |
| | OT [50] | $0.755_{\pm0.06}$ | $0.622_{\pm0.09}$ | $0.603_{\pm0.04}$ | $0.747_{\pm0.08}$ | $0.695_{\pm0.09}$ |
| | PANTHER [63] | $0.758_{\pm0.06}$ | $0.665_{\pm0.10}$ | $0.612_{\pm0.07}$ | $0.757_{\pm0.10}$ | $0.716_{\pm0.10}$ |
| **MIL** (supervised. UNI) | AttnMISL [81] | $0.627_{\pm0.08}$ | $0.639_{\pm0.10}$ | $0.485_{\pm0.06}$ | $0.581_{\pm0.12}$ | $0.649_{\pm0.09}$ |
| | ILRA [77] | $0.649_{\pm0.10}$ | $0.555_{\pm0.10}$ | $0.550_{\pm0.04}$ | $0.632_{\pm0.02}$ | $0.637_{\pm0.14}$ |
| | TransMIL [61] | $0.612_{\pm0.07}$ | $\mathbf{0.684_{\pm0.06}}$ | $0.595_{\pm0.06}$ | $0.695_{\pm0.08}$ | $0.671_{\pm0.10}$ |
| | ABMIL [31] † | $0.644_{\pm0.05}$ | $0.608_{\pm0.09}$ | $0.550_{\pm0.06}$ | $0.669_{\pm0.07}$ | $0.684_{\pm0.06}$ |
| | ABMIL reproduce | $0.633_{\pm0.06}$ | $0.612_{\pm0.08}$ | $0.540_{\pm0.07}$ | $0.671_{\pm0.08}$ | $0.691_{\pm0.08}$ |
| | + PathVQ | $0.655_{\pm0.05}$ | $0.649_{\pm0.12}$ | $\underline{0.608_{\pm0.05}}$ | $0.721_{\pm0.10}$ | $0.760_{\pm0.08}$ |
| | Δ over ABMIL | 2.2% ↑ | 3.7% ↑ | 6.8% ↑ | 5.0% ↑ | 6.9% ↑ |
| | +PathVQ + SPT | $\mathbf{0.674_{\pm0.06}}$ | $0.659_{\pm0.11}$ | $\mathbf{0.616_{\pm0.05}}$ | $\mathbf{0.748_{\pm0.11}}$ | $\mathbf{0.778_{\pm0.08}}$ |
| | Roformer | $0.602_{\pm0.09}$ | $0.617_{\pm0.13}$ | $0.572_{\pm0.07}$ | $0.721_{\pm0.08}$ | $0.655_{\pm0.13}$ |
| | +PathVQ | $0.644_{\pm0.07}$ | $0.587_{\pm0.09}$ | $0.597_{\pm0.05}$ | $\underline{0.741_{\pm0.09}}$ | $0.748_{\pm0.09}$ |
| | +PathVQ + SPT | $0.673_{\pm0.07}$ | $0.679_{\pm0.08}$ | $0.603_{\pm0.05}$ | $0.734_{\pm0.11}$ | $\underline{0.765_{\pm0.08}}$ |
| UNI-2 | ABMIL | $0.614_{\pm0.02}$ | $0.618_{\pm0.11}$ | $0.539_{\pm0.08}$ | $0.672_{\pm0.08}$ | $0.659_{\pm0.11}$ |
| **Slide-FMs** (SOTA, ckpt-only) | CHIEF [76] | $0.737_{\pm0.04}$ | $0.680_{\pm0.08}$ | $0.599_{\pm0.02}$ | $0.758_{\pm0.10}$ | $0.736_{\pm0.06}$ |
| | GigaPath [79] | $0.687_{\pm0.08}$ | $0.628_{\pm0.08}$ | $0.589_{\pm0.05}$ | $0.779_{\pm0.10}$ | $0.751_{\pm0.07}$ |
| | TITAN [16] (cox loss) | $0.713_{\pm0.04}$ | $0.710_{\pm0.11}$ | $0.657_{\pm0.05}$ | $0.789_{\pm0.09}$ | $0.774_{\pm0.06}$ |

We evaluate survival prediction on five TCGA datasets: BRCA, BLCA, CRC, UCEC, and KIRC. The model is trained using the negative log-likelihood (NLL, notice that some compared models' are trained via Cox-loss generally gain better result, please check Appendix A.6 for details.) loss and evaluated using the c-index with 5-fold cross validation (the result of last epoch is reported).

For fair comparison, we follow the default training pipeline of PANTHER [63], including hyper-parameters and data splits, and integrate our proposed model modifications along with pretrained weights.

**Compared Baselines:** We categorize the baselines into three groups: **Unsupervised Prototype-Based Approaches:** H2T [73] (clusters tile embeddings and pools them within each cluster), OT [50] (aggregates patch features into a set of prototypes using Optimal Transport), PANTHER [63] (models prototype tile embeddings via a Gaussian Mixture Model). **Supervised MIL Models:** AttnMISL [81] (combines prototype-based learning with MIL), ABMIL [31], TransMIL [61], ILRA [77], and Transformer with RoPE [64]. **Slide-Level FMs:** CHIEF [76]: A large-scale ABMIL-pretrained model using contrastive learning to predict organ source, with CTransPath as the tile feature extractor (mean-pooled features), GigaPath [79] and TITAN [17]. UNI-2 [10] feature extractor with ABMIL model.

**Survival Prediction Results Analysis:** The results are reported in Table 2. We can observe that ABMIL and Roformer show significant improvement when combined with our PathVQ compressor into UNI. But for Roformer with large-scale of parameters, the performance get easily overfitting to labels thus showing interior performance, which demonstrate the necessity of pretraining. For UNI-2, the improvement is marginal, proving our previous claim on the [CLS] bottleneck of scalability.

### 3.3 Ablations and Visualization

**Performance vs. Complexity across reduction methods** is shown in Figure 5. PCA: Reduces token dimensions by projecting them onto a low-rank linear subspace, preserving variance but potentially discarding discriminative details. The F1-score reaches about 0.70.

Average Pooling: Aggregates tokens by computing their mean, leading to compact representations but often oversmoothing critical local information. We have conducted pooling original 196=14x14 token features into 4x4 and 7x7 tokens, resulting in about 10 and 2 times token reduction. However, the result only show similar performance compared to original CLS token.

CLS and overall average-pooling: as performed in virchow, the CLS and average-pooling token can complement a little to each other and resulting in around 1.0 point improvement.

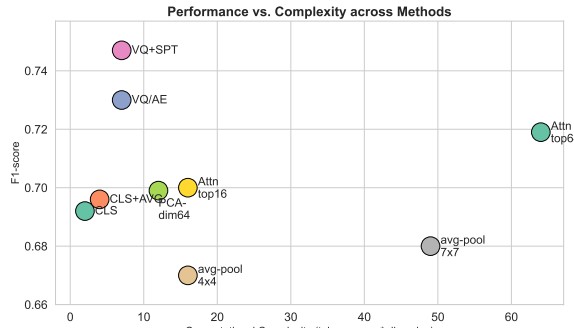

Figure 5: Performance–complexity trade-off across various token reduction strategies on BRACS. Our proposed VQ method achieves the best balance between accuracy and efficiency, delivering the highest F1-score (0.73) with significantly reduced token complexity. Other approaches, including PCA, average pooling, and CLS-based methods, show varying trade-offs.

We also performed token selection via CLS token attention top-k ranking. The selected top64 can reach a F1-score around 0.72 but is still too heavy in computation cost. The selected top16 can reach a F1-score of 0.70, can be seen as a trade-off, but lose too much on performance.

Our proposed VQ-based method achieves the best balance, attaining the highest F1-score of 0.73 while significantly reducing token complexity. In contrast, other strategies demonstrate varying degrees of compromise between accuracy and computational cost.

**Convs pretraining ablation:** Please check Appendix Figure 6.

For additional ablation studies and visualizations, please refer to the Appendix A.4. The main findings: The VQ reconstruction performance remains relatively stable when varying the quantized embedding dimension (32) and codebook size (16384). The reconstruction results of MSVQ show improvements.

## 4 Conclusion and Limitations

In this work, we introduced a novel vector quantization (VQ) distillation framework to address the inherent bottleneck of existing computational pathology foundation models in whole-slide image analysis. Furthermore, our multi-scale VQ strategy unifies patch- and tile-level features, not only improving feature reconstruction but also serving as an effective self-supervised learning supervision target for slide-level pretraining. The main **limitation** is that the VQ learning process need extra training data. And the VQ still lose some information though it is acceptable. By efficiently compressing patch-level spatial tokens while preserving critical spatial and contextual information, our method significantly reduces storage and computational costs without compromising performance.

## 5 Acknowledgments

This study was partially supported by "Pioneer" and "Leading Goose" R&D Program of Zhejiang (Grant 2025SDXHDX0003), the National Natural Science Foundation of China (Grant No.62506306), and foundation of Muyuan Laboratory (Program ID: 14106022401,14106022402). Furthermore, essential technical support was provided by the D-PathAI platform, including its hardware and software, which was developed by Hangzhou Dipath Technology Co., Ltd.

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

# A  Supplementary Materials

## A.1  Related Work

### A.1.1  Pathological Whole Slide Image Analysis

Whole Slide Images (WSIs) contain a wealth of visual information that plays a crucial role in pathological analysis [5, 49]. However, obtaining detailed cell-level annotations is both labor-intensive and time-consuming [5, 49, 9], posing a significant challenge for large-scale WSI analysis. To address this, weakly-supervised learning has emerged as a promising direction in computational pathology.

**Computational Pathology Foundation Models** The performance of early WSI MIL approaches [31, 11, 39, 49, 84, 34, 56, 14, 1, 9, 61, 41, 11, 44, 23, 7, 21] relied heavily on tile-level features extracted from pre-trained models [42, 20]. To address this, FMs have been developed and shown significant advancements in both tile-level [75, 48, 10, 20, 72, 59] and WSI-level analysis [79, 17]. These FMs leverage visual Self-supervised Learning (SSL) techniques [6, 52, 13, 12] on large-scale unlabeled datasets [75, 10, 20] or organize pathology image-text pairs to learn multimodal representation [57, 17, 48]. FMs have demonstrated superior performance in downstream tasks such as cancer subtyping, survival prediction, and biomarker identification.

Recently, authors in Hest-1k [32] observe that tile encoders like Conch [48] can be further fine-tuned to obtain better downstream tile task result. However, the key challenge that the computational cost high-resolution WSIs makes fine-tuning [85, 42] FMs with overwhelm parameters difficult. Most approaches [48, 10, 20] resort to using pretrained [CLS] token representation of tile-level FM as slide-level inputs, which may lead to the loss of critical spatial information. Some models, such as UNI-2 [10], attempt to scale up ViTs into larger-size as tile encoders to extract better feature representations, but only achieve marginal improvements from ViT-Large to ViT-Giant. We argue that this performance bottleneck stems from the spatial information loss inherent in [CLS] token representations. Other efforts, such as Virchow-2 [90], find that combining [CLS] tokens with [AVG] (average pooling of all spatial tokens) can yield some improvement (less than 1 point). Motivated by these findings, we propose to keep but compress all spatial patch tokens and further extract useful information for downstream WSI tasks analysis.

**Slide-level FMs / SSL pretraining:** Some recent slide-level FMs, e.g. GigaPath [79] and TITAN [17] are modeled via Transformer with 6 to 12 layers, then pretrained via MAE [25] and iBOT [88] respectively. Some other slide SSL models [26, 37, 9] also propose to employ slide-level augmentation with contrastive learning (CL) [12, 52] for pretraining. But there are some training problems of these works: 1) The main augmentations in slide-level to generate different views for CL is limited, like crop or random-drop, since the tile feature are pre-processed and stored. This hinders the performance of CL pretraining. 2) The self-supervised target. The iBOT, [88] used in TITAN [17] predicting masked token to match online-tokenizer, is not so stable during training. The MAE in GigaPath [79] need to regress the feature of masked tiles, which may be too difficult to fit and hinder downstream tasks. The CHIEF [76], on the other way, pretrain ABMIL by constrastively predicting tumor organ source (extra information).

Different to these work, in this paper we train a offline tile tokenizer via vector quantization which can offer self-supervision for both WSI-Transformer mask modeling and ABMIL. This is more stable for pretraining and need no further information.

### A.1.2  Vector Quantization

Vector quantization (VQ) [22] is a fundamental technique in signal processing and machine learning, widely used for data compression, clustering, and generative modeling [18, 71, 58]. Recent developments in deep learning have led to neural vector quantization methods, such as Vector Quantized Variational Autoencoders (VQ-VAEs) [71, 58, 8] and quantized transformers [46], which integrate VQ into end-to-end learning pipelines to enhance expressiveness and efficiency. To further improve VQ, techniques such as residual quantization [38] and rotation tricks [19] have been proposed. Recent studies [82, 83] reveal that lower-dimensional quantized vectors (dimension size ranging from 8 to 32) can improve codebook usage and reconstruction performance, providing strong compression capabilities that benefit this study. Unlike recent VQ methods that focus primarily on visual genera-

tion [71, 68, 58], our work focus on feature compression and distillation of pretrained pathology tile encoder features via a VQ quantizer.

## A.2 Data Description

BReAst Carcinoma Subtyping (BRACS) [4] collect H&E stained Histology Images, containing 547 WSIs for three lesion types, i.e., benign, malignant and atypical, which are further subtyped into seven categories. Here, since the WSIs number is limited, we only perform three class subtyping.

TCGA [69]: Breast Invasive Carcinoma (BRCA, n = 1, 041, WSI = 1, 111), Colon and Rectum Adenocarcinoma (CRC, n = 566, WSI = 575), Bladder Urothelial Carcinoma (BLCA, n = 373, WSI = 437), Uterine corpus endometrial carcinoma (UCEC, n = 504, WSI = 565), Kidney renal clear cell carcinoma (KIRC, n = 511, WSI = 517), Brain Lower Grade Glioma (LGG) and Glioblastoma Multiforme (GBM) constitute WSI = 463. The train/val split is performed on the patient level.

## A.3 Experimental settings

**VQ Pretraining:** We conduct VQ pretraining on 1M randomly cropped $224 \times 224$ tiles extracted from all TCGA [69] diagnostic pathology WSIs. During training, the FM backbone (e.g., UNI with ViT-Large) remains frozen. The input tile images are augmented using `RandomCrop` (minimum ratio: 0.4) and `RandomHorizontalFlip` (probability: 0.5). The codebook has a size of $C = 8192$ with an embedding dimension of 16. The MLP encoder consists of two linear layers with a `tanh` activation in between, transforming the feature dimension from 1024 to 16. The decoder first upsamples the feature dimension from 16 to 768 using a linear layer, followed by three Transformer blocks. Another linear layer then maps the features from 768 to 1024, ensuring alignment with the original feature tokens.

For MSVQ, we employ a multi-scale resolution list: $\{1 \times 1, 2 \times 2, 4 \times 4, 7 \times 7, 14 \times 14\}$.

The model is trained on 4 RTX-3090 GPUs for 50 epochs using a batch size of 128 tile images. The total training time is approximately 22 hours. The learning rate is set to $2 \times 10^{-4}$ with a 5-epoch warmup, followed by cosine decay to a minimum learning rate of $1 \times 10^{-5}$. The weight decay is $1 \times 10^{-4}$, and the AdamW optimizer is used with $\beta$ parameters set to $(0.9, 0.99)$.

**WSI-SSL Pretraining:** We crop all TCGA diagnostic WSIs into regions of resolution $3584 \times 3584$, yielding a dataset of approximately 250k regions. To facilitate SSL, a pretrained MSVQ model is used to extract the quantized indices of each tile within a region, requiring only about 65MB for storage.

All pretraining is conducted on 4 GPUs for 20 epochs with an initial learning rate of $5 \times 10^{-4}$. The first 2 epochs serve as a warmup phase, followed by cosine decay to a minimum learning rate of $1 \times 10^{-5}$. The AdamW optimizer is employed with $\beta$ parameters set to $(0.9, 0.98)$ to ensure fast convergence.

For ABMIL pretraining, a batch size of 64 is used due to the model's simplicity. For WSI-Transformer, the batch size is set to 32, with 96 masked tokens out of 256. The learning objective for both models is formulated as a cross-entropy loss over 8192 categories, with ABMIL additionally utilizing soft targets.

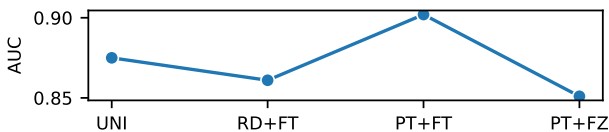

Figure 6: The PathVQ feature with Convs need pretraining and aligning tile's level-0 feature to attain good feature space and compression. RD: random initialize Convs. FT: fine-tuning during WSI analysis. PT: pretrained during VQ learning (align to level-0 tile feature). FZ: freeze during WSI analysis.

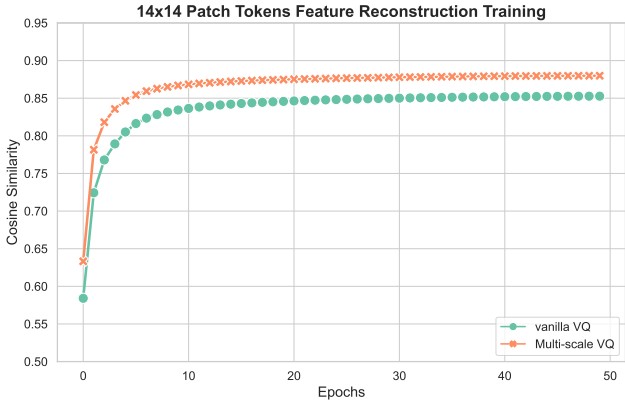

Figure 7: Reconstruction using Multi-Scale (MSVQ) or not. MSVQ obviously improve the rec performance.

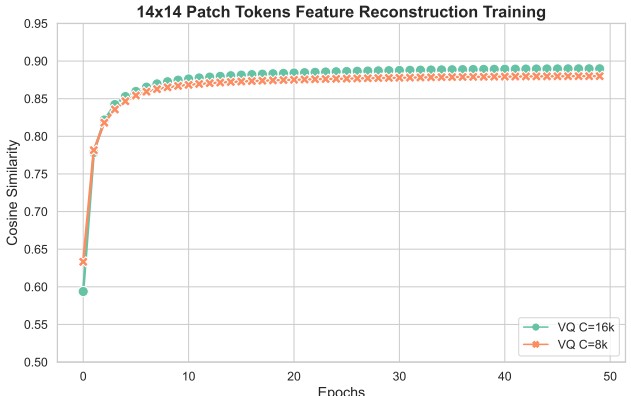

Figure 8: Reconstruction ablation on quantization codebook size, 8k and 16k.

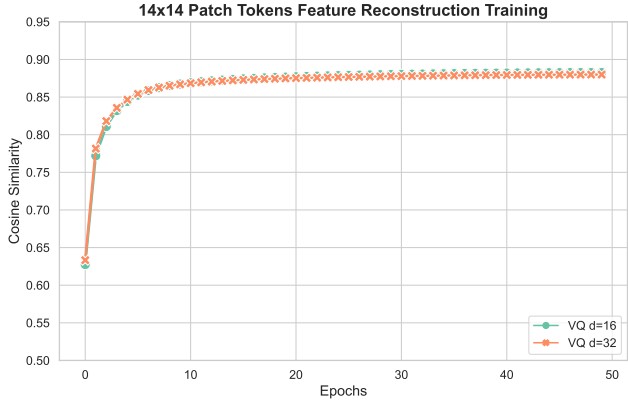

Figure 9: Reconstruction ablation on quantization codebook embedding, 16 and 32.

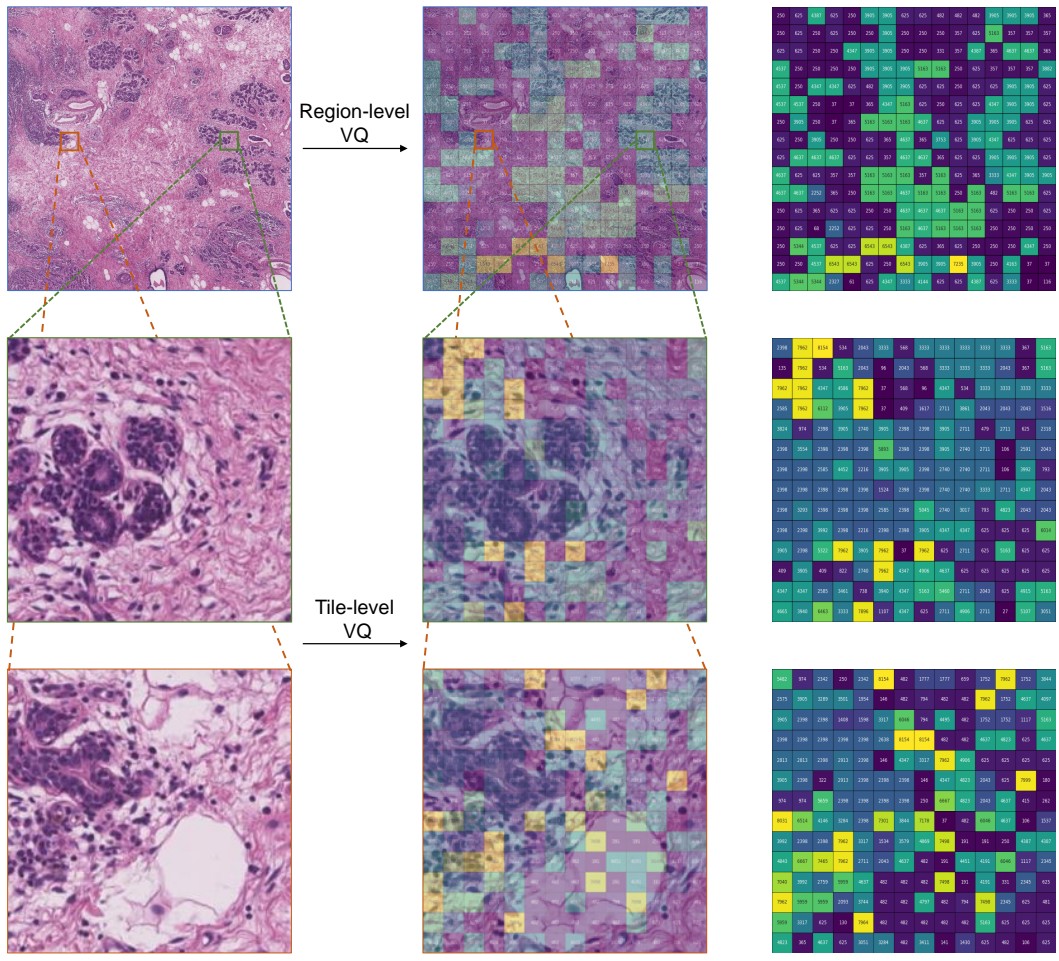

Figure 10: The VQ are performed on both tile-level and patch-level. The quantized index can be seen as a type of prototypes (n=8k) with strong interpretability. Be aware that since the codebook is too large, the heatmap on different index may share similar color.

## A.4 Ablations on VQ

**Multi-Scale (MSVQ)** Please check Figure 7.

**Codebook Size** Please check Figure 8.

**Codebook Embedding Dimension** Please check Figure 9.

## A.5 Illustrative Visualization

Please check Figure 10.

## A.6 Survival Prediction Loss Comparison

Most current MIL methods [31, 61] use NLL loss since WSI batch size is limited (most settings are 1) when the MIL models need fine-tuned with large bag size (GPU memory limit). However, the NLL loss is not optimal for this setting [60]. Cox loss [60], on the other hand, is better than NLL in performance but it need large batch size to calculate the hazard ranking matrix among different samples, which is currently inevitable for MIL models. However, this is easy to be implemented if the bag-size is small (e.g. using unsupervised learning) to prototyping instances' features like PANTHER [63]. Or using a strong WSI pretrained module like TITAN [17] pre-extract the slide-level representations. Though currently our method can not surpass above methods, but it show strong

improvement compared to other baselines which also using NLL loss. And we will further explore this problem (combining Cox loss into fine-tuning WSI-level representation model) in the future.

