# OpenReview forum: "PathVQ: Reforming Computational Pathology Foundation Model for Whole Slide Image Analysis via Vector Quantization"
_NeurIPS.cc/2025/Conference — NeurIPS 2025 poster_

### Official Review · Reviewer_nSUP · 2025-06-30

**Clarity:** 2
**Significance:** 3
**Originality:** 3
**Rating:** 4
**Confidence:** 4

**Summary:**

This paper focuses on improving existing ViT-based pathology foundation models (FMs). Most current approaches represent each patch using a [CLS] token and aggregate multiple [CLS] tokens to obtain the representation of a WSI. In contrast, this work proposes a novel method that distills the tokens of each patch onto a codebook using vector quantization (VQ). This reduces the original tile representation from $n \times D$ to $C \times d$, achieving a trade-off between computational cost and information preservation.

**Questions:**

1, How does PathVQ perform on more foundation models?

2, How does the training cost of PathVQ compare to fine-tuning?

3, What is the effectiveness of MSVQ? Are there ablation studies?

**Ethical Concerns:**

["NO or VERY MINOR ethics concerns only"]

**Final Justification:**

My major concerns regarding the writing and experimental settings have been addressed. I appreciate the idea of enhancing foundation models and will raise my score accordingly.

**Limitations:**

yes

**Quality:**

2

**Strengths And Weaknesses:**

Strengths

- The paper provides an accurate summary of existing ViT-based FMs in computational pathology.
- The idea of post-processing the outputs of large-scale models is interesting and potentially useful for improving efficiency.

Weaknesses

1. Figure 1 is confusing. The meaning of "attntop64/16" is unclear. I assume the left vertical axis shows the cosine similarity to measure information reconstruction (possibly reflecting the preservation of spatial information), in which the proposed VQ method performs well. However, the memory consumption on the right appears suboptimal compared to other methods. Moreover, it's unclear how well the proposed method performs on downstream tasks and whether it generalizes across different foundation models.

2. Lack of detail in the MSVQ description. The paper does not sufficiently explain how aggregation is done across different scales, nor does it clearly describe the final output structure. This section is hard to follow.
Additionally, the pseudocode in Algorithm 1 is difficult to interpret due to undefined variables and functions. A more precise definition and breakdown are needed. Ablation studies related to this part are also missing.

3. Insufficient experimental validation for VQ. From Figure 4, it seems that VQ is only applied to UNI tokens, which is not a particularly resource-intensive operation. The study should have evaluated more commonly used foundation models and included more patch-level datasets to better validate the generalizability of the VQ approach.

4. Unclear comparison in training efficiency. Based on Figure 4, the PathVQ method is compared against fine-tuning the last block of the UNI model. While fine-tuning performs slightly better on CRC and VQ is slightly better on C17, the paper does not report training or computational cost, making it hard to assess the efficiency gains of PathVQ over conventional fine-tuning. Similarly, in the slide-level classification tasks, it is unclear why the authors did not report PathVQ's performance on larger datasets like Gigapath and TITAN.

---

> ### Author Rebuttal · Authors · 2025-07-30
>
> We thank the reviewer for the insightful comments and appreciate the reviewer's concerns. We hope the reviewer can re-evaluating our paper based on this response.
>
>
> >W1: Figure 1 is confusing. The meaning of "attntop64/16" is unclear. I assume the left vertical axis shows the cosine similarity to measure information reconstruction (possibly reflecting the preservation of spatial information), in which the proposed VQ method performs well. However, the memory consumption on the right appears suboptimal compared to other methods. Moreover, it's unclear how well the proposed method performs on downstream tasks and whether it generalizes across different foundation models.
>
> Figure 1 is a intuitive illustration of our motivation on why we need use more spatial patch tokens to replace CLS. The attn-top64/16 indicates another aspect/baseline by using less tokens (token reduction, implemented by ranking the CLS token attention weight on other patch tokens, we select the top64 and 16 most attented tokens) compared to ours less dimension.
> The left vertical axis shows the similarity of information preserving of the curves. The **right vertical axis** show the memory/computational cost of **the bubbles**. So **the memory consumption on the right appears suboptimal compared to other methods** is a mis-understanding. As shown by the position of bubbles, our method only need a few extra cost compared to CLS but much better than token reduction. In addition, the bubble size indicates the downstream tasks performance (BRACS AUC), as explained in Figure 1 caption.
>
> We feel sorry about the confusing and will make it more clear in next version. For different foundation models, it may make the figure more confusing, so we will included it in appendix in next version, some of its experimental ablations are included in following content.
>
> >W2: Lack of detail in the MSVQ description. The paper does not sufficiently explain how aggregation is done across different scales, nor does it clearly describe the final output structure. This section is hard to follow. Additionally, the pseudocode in Algorithm 1 is difficult to interpret due to undefined variables and functions. A more precise definition and breakdown are needed. Ablation studies related to this part are also missing.
>
> The description on MSVQ can be found in line 139-162 and the figure 2. Sorry for the unclear elaboration. Here is a brief understanding of MSVQ in two-scale (tile-level and patch-level): 1) the tile-level VQ first encodes the average-pooling of 14x14=196 tokens (denoted as AVG, one token can be seem as same to CLS token). 2) upsampling the AVG VQ feature back to 14x14,  calculate the residual (difference) between the the upsampled feature map and original patch-level feature map, then the quantization is performed on the residuals. This can be further extened into multi-scale (1x1, 2x2, ..., 14x14). We will make it more clear in next version including a more precise definition and breakdown on algorithm, figure and text.
>
> For ablation study, we have showed the VQ learning cos-similarity on patch-scale only and multi-scale in Figure 7 (page 18).
> For more scales ablation, we here show the cos-similarity and downstream tasks performance comparison:
> | Method                                   | Cos-similarity | F1-score |
> |------------------------------------------|----------------|----------|
> | 14×14 VQ                                 | 0.851          | 0.720    |
> | 1×1 + 14×14 MSVQ                         | 0.859          | 0.726    |
> | 1×1 + 2×2 + 14×14 MSVQ                   | 0.872          | 0.725    |
> | 1×1 + 2×2 + 4×4 + 7×7 + 14×14 MSVQ       | 0.881          | 0.730    |
> | MSVQ + SPT                               | 0.881          | 0.747    |
>
>
> >W3 & Q1: Insufficient experimental validation for VQ. From Figure 4, it seems that VQ is only applied to UNI tokens, which is not a particularly resource-intensive operation. The study should have evaluated more commonly used foundation models and included more patch-level datasets to better validate the generalizability of the VQ approach.
>
> We here list the performance of implementing VQ  with Conch[1] considering it is trained via image-text constrastive learning aligment, thus may showing some difference to image-only SSL like UNI/Gigapath/Virchow, etc. Based on  the result, we find that we can get similar performance gains from PathVQ. In next version, we will inlcude more Path foundation models and its evaluation on more dataset when combined with our VQ methods.
>
> | Method                                   | AUC | F1-score |
> |------------------------------------------|----------------|----------|
> | UNI+ABMIL                                 | 0.875          | 0.692    |
> | UNI+PathVQ+ABMIL                        | 0.902          | 0.730    |
> | UNI+PathVQ+ABMIL+SPT                        | 0.906          | 0.747    |
> | Conch+ABMIL                                 | 0.872          | 0.683    |
> | Conch+PathVQ+ABMIL                        | 0.897          | 0.718    |
> | Conch+PathVQ+ABMIL+SPT                        | 0.910          | 0.751    |
>
>
> >W4.1 & Q2: Unclear comparison in training efficiency. Based on Figure 4, the PathVQ method is compared against fine-tuning the last block of the UNI model. While fine-tuning performs slightly better on CRC and VQ is slightly better on C17, the paper does not report training or computational cost, making it hard to assess the efficiency gains of PathVQ over conventional fine-tuning.
>
> Dispite the VQ pretraining cost, the overall fine-tuning cost of PathVQ compared to fine-tuning is slight. For slide-encoder inference, the storage cost is about 4 times and the training cost is about 2~3 times. However, during the overall WSI inference after model deployment in clinical diagnosis, the tile-level encoder inference takes most time cost. Generally, the tile-level encoder takes about 2 minites but the slide encoder only takes less than 1 seconds (ABMIL, using VQ takes about 2 seconds) and 3 seconds for slide-transformer while using VQ takes about 5 seconds.
>
> Here we list the training cost of 1 epoch for training BRACS, the main training speed is bottlenecked by the data loading during 2-stage training, so it does not impact too much during inference.
>
> | Aspect |PathVQ|Baseline |
> | - | -|-|
> | Fine-tuning Cost of BRACS 1 epoch  | ~28 seconds | ~13 seconds       |
> |Storage Cost (Slide Encoder)     | \~4×   | 1   |
> | Training Cost (Slide Encoder)   | \~2–3×   |1    |
> | Inference Time (Tile Encoder)      | \~1-2 minutes   | Dominates WSI inference time     |
> |Inference Time (Slide Encoder)     | ~2s (ABMIL + VQ)      | <1 second (ABMIL),  Slide-level inference is fast|
> | Inference Time (Slide Transformer) | ~3s (with VQ)  | 1s (w/o VQ)   |
>
> For tile/RoI level experiments, the main purpose is to show that our PathVQ can attain most information of original tokens, thus may showing similar performance to end-to-end fine-tuning considering that the foundation models' ability. Since end-to-end can not be performed in slide-level, the PathVQ or preserving all generated patch tokens may strongly improve the WSI performance compared to end-to-end learning.
>
> >W4.2 Similarly, in the slide-level classification tasks, it is unclear why the authors did not report PathVQ's performance on larger datasets like Gigapath and TITAN.
>
> For mroe dataset evaluation like TITAN, we have conducted about 7 WSI level and 2 RoI level in this paper, I believe it is sufficient to demonstrate the effectiveness of our method. We here include 2 extra slide tumor classification task (for rebuttal length) and will validate more in next version.
>
> | Method | BCNB-ER | TCGA-LUAD TP53 |
> |-|-|-|
> |-|AUC|AUC|
> | UNI+ABMIL | 0.867          | 0.758    |
> | UNI+PathVQ+ABMIL                        | 0.891          | 0.775    |
>
> >Q3: What is the effectiveness of MSVQ? Are there ablation studies?
>
> For MSVQ ablation, please see response in W2.
>
> For MSVQ novelty perspective, our work builds upon established techniques such as VQ and SSL. Our core contribution lies in **unifying two longstanding challenges in WSI foundation model pretraining** through a coherent and scalable **multi-scale VQ framework**:
>
> **(1) Addressing the CLS token bottleneck via patch-level (16x16) VQ:**
>
> CLS token–based representations often discard fine-grained spatial information, leading to suboptimal performance on downstream pathology tasks. To mitigate this, we apply patch-level (patch size = 16) VQ/autoencoding to preserve local semantics while significantly reducing spatial dimensionality. This allows us to compress the token space without resorting to a single global embedding.
>
> **(2) Improving the SPT (Slide-level Pretraining) objective via tile-level (224x224) VQ:**
>
> ..Just like MAE learn too many low-frequency feature of raw-pixel image, directly using feature-level targets also not align well with task-specific semantics.
>
> To resolve this, we adopt a **tile-level (tile size = 224) VQ tokenizer** that discretizes feature targets into informative codebook entries. This approach enables more stable and effective slide-level representation learning compared to raw pixel or continuous-feature reconstruction.
>
> **Unified framework via MSVQ:**
>
> Rather than treating patch-level compression and slide-level representation learning as disjoint processes, we introduce a **MSVQ** that links both levels: patch-level VQ captures local detail, while tile-level VQ supports robust and interpretable global objectives. This unified view of representation compression and learning makes our approach both practically scalable and theoretically grounded for WSI tasks.

---

> > ### Comment · Reviewer_nSUP · 2025-08-06
> > **good rebuttal**
> >
> > Thank the authors for the rebuttal. My major concerns regarding the writing and experimental settings have been addressed.
> >
> > I recommend incorporating the clarifications provided in the rebuttal into the final version, especially to improve the clarity of the introduction section.
> >
> > I appreciate the idea of enhancing foundation models and will raise my score accordingly.

---

> > > ### Author Response · Authors · 2025-08-07
> > > **Thanks**
> > >
> > > Thank you very much for your time and for raising your score. We are glad that our response and detailed analysis addressed your concerns.
> > > We will carefully revise the manuscript according to all the valuable feedback received during this review process.
> > >
> > > Thank you again for your support.

---

### Official Review · Reviewer_udpH · 2025-07-01

**Clarity:** 3
**Significance:** 3
**Originality:** 2
**Rating:** 4
**Confidence:** 4

**Summary:**

The authors propose PathVQ, a vector quantization-based framework for compressing spatial patch tokens in WSI analysis. To reduce computational burden while preserving rich contextual information, they introduce a multi-scale VQ strategy and apply it in a SSL setting to support scalable pretraining and downstream tasks such as classification and survival analysis. Results show improved performance over existing aggregation methods across several benchmarks.

**Questions:**

Please address the concerns above.

**Ethical Concerns:**

["NO or VERY MINOR ethics concerns only"]

**Final Justification:**

The authors have sufficiently addressed my concerns regarding model motivation and evaluation.

**Limitations:**

Yes

**Paper Formatting Concerns:**

No concerns

**Quality:**

3

**Strengths And Weaknesses:**

Strengths:
- The paper addresses patch token aggregation and model throughput, important problems in CPATH and working with WSI foundation models
- Experiments cover multiple downstream tasks, including survival, and include comparisons with recent MIL, prototype based, and FMs

Weakness + questions:
- Using vector quantization to resolve the trade off between using CLS and patch token is not well justified. While vector quantization compresses patch feature dimensions, the amount of memory needed still scales linearly to the number of patch tokens - instead of using CLS, which would be constant. Other simpler methods, such as using a linear projector, could achieve similar memory savings while maintaining similar model performances.
- The motivation behind using SSL with VQ is unclear. How does the model perform with SPT without vector quantization?
- The effect of multi-scale VQ is also unclear. How does the number of scales and their resolution affect the model performance?
- It is concerning that PathVQ underperforms prototype/slide-FM based methods available, especially in survival prediction, which are the more challenging and complex down-stream tasks

---

> ### Author Rebuttal · Authors · 2025-07-30
>
> We thank the reviewer for the insightful comments and appreciate the reviewer's concerns. We hope the reviewer can re-evaluating our paper based on this response.
>
> >W1.1: Using vector quantization to resolve the trade off between using CLS and patch token is not well justified. While vector quantization compresses patch feature dimensions, the amount of memory needed still scales linearly to the number of patch tokens - instead of using CLS, which would be constant.
>
> We have compressed the 196 tokens with d=1024 into d=16 / d=32, the cost comparison to CLS token is 196*16/ (1*1024)=3.0625, which is acceptable. In our experiments, we find that 2~16x input cost compared to CLS are acceptable in training GPU memory cost and latency.
>
> As for token scales problem, there may be some mis-understanding. We do not input all tokens into ABMIL or Transformer, we utilize a upsampling layer (like conv with 4 stride size) to convert the feature into dim 1024 or 2048, this resulted feature is of high rank, make it more easily for down-stream task anaysis.
>
> we tackle a fundamental limitation in transformer-based slide-level models: **insufficient input rank**, which constrains their capacity to represent complex WSI data. As noted in [2], the rank of the attention matrix in a standard transformer is theoretically bounded:
>
> $$
> \text{rank}(QK^\top) \leq \min(\text{rank}(Q), \text{rank}(K)) = \min(n, d),
> $$
> where n is the sequence length and d is the feature dimension. When using a single CLS token (or averaged tile-level CLS features), the actual rank is severely limited. Though in [2] authors try to improve the rank via local attention and positional embedding, the input rank limitation still exists. For example, we observe that in **Conch**, the effective rank of WSI embeddings using tile-level CLS is approximately **50**, and in **UNI**, it is around **400**—both significantly lower than the typical WSI sequence length (often exceeding **5,000** tokens).
>
> | Model | Rank (Mean ± Std) | WSI Length (Mean ± Std) | AUC Transformer |
> | --- | --- | --- | --- |
> | **Conch** | 53.6 ± 6.2 | 3356.1 ± 1840.6 | 0.861 |
> | **UNI** | 367.3 ± 109.7 | 3356.1 ± 1840.6 | 0.882 |
> | **Ours** | 765.4 ± 88.3 | 3356.1 ± 1840.6 | 0.910 |
>
> In contrast, our method reconstructs spatial tokens via VQ-based representations, enabling **a much higher-rank representation** of the WSI content. We can also **increase the feature dimensionality (e.g., to 2048)**, which is more effective than simply up-projecting CLS features from 1024 using a linear layer.
> For context, in standard ViT-Large (e.g., ViT-L/16 with 224×224 inputs), a 1024-d embedding is used for just 196 tokens. Large language models (e.g., LLaMA, Qwen ranging from 3~7B) employ embeddings of 2048–4,096 dimensions to represent typical question lengths. By analogy, for long-sequence AI applications such as WSIs, **our approach provides a scalable and rank-preserving alternative perspective**, and may generalize to other domains facing similar sequence-length bottlenecks.
>
>
>
>
> >W1.2 Other simpler methods, such as using a linear projector, could achieve similar memory savings while maintaining similar model performances.
>
> Using AE (without VQ), even PCA compression  can obtain similar result, as showing in Figure 5, but our VQ goal is not only for compression. While our work builds upon established techniques such as VQ and SSL, our core contribution lies in **unifying two longstanding challenges in WSI foundation model pretraining** through a coherent and scalable **multi-scale VQ framework**:
>
> **(1) Addressing the CLS token bottleneck via patch-level (16x16) VQ:**
>
> CLS token–based representations often discard fine-grained spatial information, leading to suboptimal performance on downstream pathology tasks. To mitigate this, we apply patch-level (patch size = 16) VQ/autoencoding to preserve local semantics while significantly reducing spatial dimensionality. This allows us to compress the token space without resorting to a single global embedding.
>
> **(2) Improving the SPT (Slide-level Pretraining) objective via tile-level (224x224) VQ:**
>
> Traditional masked autoencoding (e.g., MAE) focuses on reconstructing pixel inputs, which is infeasible for WSI-scale images due to GPU memory limits. We instead mask and predict **intermediate tile-level feature representations**, but just like MAE learn too many low-frequency feature of raw-pixel image [1], directly using feature-level targets also not align well with task-specific semantics.
>
> To resolve this, we adopt a **tile-level (tile size = 224) VQ tokenizer** that discretizes feature targets into informative codebook entries. This approach enables more stable and effective slide-level representation learning compared to raw pixel or continuous-feature reconstruction.
>
> **Unified framework via Multi-Scale VQ:**
>
> Rather than treating patch-level compression and slide-level representation learning as disjoint processes, we introduce a **Multi-Scale VQ architecture** that links both levels: patch-level VQ captures local detail, while tile-level VQ supports robust and interpretable global objectives. This unified view of representation compression and learning makes our approach both practically scalable and theoretically grounded for WSI tasks.
>
>
> >W2: The motivation behind using SSL with VQ is unclear. How does the model perform with SPT without vector quantization?
>
> The motivation behind using SSL with VQ is elaborated above. We did make some experiment on SPT without VQ but find limited performance. One of the main reason is that direct predict the input feature space may not benifit downstream task (just like MAE predict raw-pixels may not fit classification best). Differently, the VQ can work as a kind of prototyping, it is inner categories of pathology images. What's more, the VQ is performed offlined, which is better compared to DINOv2 using online tokenizer for categories/index prediction.
>
> | Method                                   | AUC | F1-score |
> |------------------------------------------|----------------|----------|
> | UNI+Transformer                                 | 0.882          | 0.678    |
> | UNI+Transformer + feature alignment SPT  | 0.894          | 0.725    |
> | UNI+Transformer+ VQ SPT                        | 0.910          | 0.754   |
>
> >W3: The effect of multi-scale VQ is also unclear. How does the number of scales and their resolution affect the model performance?
>
> Despite the unified goal (feature compression and SSL objective) of MSVQ, the MSVQ can enhance the VQ recovering performance since the image are borned with multi-scale characteristic.
> For ablation study, we have showed the VQ learning cos-similarity on patch-scale only and multi-scale in Figure 7 (page 18).
> For more scales ablation, we here show the cos-similarity and downstream tasks performance comparison:
> | Method                                   | Cos-similarity | F1-score |
> |------------------------------------------|----------------|----------|
> | 14×14 VQ                                 | 0.851          | 0.720    |
> | 1×1 + 14×14 MSVQ                         | 0.859          | 0.726    |
> | 1×1 + 2×2 + 14×14 MSVQ                   | 0.872          | 0.725    |
> | 1×1 + 2×2 + 4×4 + 7×7 + 14×14 MSVQ       | 0.881          | 0.730    |
> | MSVQ + SPT                               | 0.881          | 0.747    |
>
>
> >W4: It is concerning that PathVQ underperforms prototype/slide-FM based methods available, especially in survival prediction, which are the more challenging and complex down-stream tasks.
>
> The comparison to slide-FM is not fair since they have used over 10 times of  prviate WSI data than ours (only public avaiable), but we can still exceed them in WSI classification tasks, which demonstrate the potential of our method, especially the feature compression has not been explored in these previous methods.
>
> For survival prediction tasks using prototyping (Panther) or slide-FM (TITAN), a key difference lies in their use of a larger training batch size and Cox loss (as noted in lines 300–301). In contrast, following most prior works such as ABMIL-based methods, we adopt a batch size of 1 and use NLL loss, which typically results in performance that is 3–5 points lower compared to settings with larger batch sizes and Cox loss.
>
> Here, for more fair comparison to Panther and TITAN, we show the comparable result with similar trainig setting (larger batchsize and cox loss):
>
> | Method  | BRCA | CRC | BLCA | UCEC | KIRC| avg|
> |- |- |-|-|-|-|-|
> | Panther (prototyping)   | 0.758          |  0.665    |  0.612|  0.757 |  0.716| 0.702 |
> | TITAN (slide-FM)     | 0.713         | 0.710    | 0.657 | 0.789|  0.774| 0.729 |
> | Ours, bs=1+ NLL loss | 0.673          | 0.679    |  0.603 |   0.734 | 0.765| 0.691 |
> | Ours, bs=16+ cox loss | 0.727          | 0.698    |  0.629 |   0.742 | 0.775| 0.714 |
>
> We would like to emphasize that survival prediction inherently exhibits an upper-bound in performance (e.g., clinicians typically achieve C-indices below 0.75 [3]). Taking into account that our model is trained on merely 1/10 of the data compared to TITAN, the achieved performance is competitive and indicates the efficiency and effectiveness of our method.
>
>
> Ref:
>
> [1] Beit: Bert pre-training of image transformers
>
> [2] Rethinking Transformer for Long Contextual Histopathology Whole Slide Image Analysis
>
> [3] Automated model versus treating physician for predicting survival time of patients with metastatic cancer.

---

> ### Comment · Reviewer_udpH · 2025-08-05
>
> I thank the authors for the detailed response. My concerns have been sufficiently addressed, and I will raise my score accordingly.

---

> > ### Author Response · Authors · 2025-08-05
> > **Thanks**
> >
> > Thanks for your time and for engaging with our rebuttal. We are sincerely grateful for your decision to raise your score. Your constructive questions and feedback have been instrumental in helping us strengthen the manuscript, and we truly appreciate your support.

---

### Official Review · Reviewer_TKaA · 2025-07-03

**Clarity:** 3
**Significance:** 2
**Originality:** 2
**Rating:** 4
**Confidence:** 4

**Summary:**

This paper proposes a novel method, PathVQ, which initially employs a feature distillation framework based on Vector Quantization (VQ) to compress spatial patch tokens into discrete indices. This achieves substantial compression (e.g., 64-fold from 1024 to 16 dimensions) while preserving high fidelity and rich spatial semantics. Subsequently, a Multi-Scale VQ (MSVQ) strategy is introduced to unify patch-level and tile-level feature quantization, thereby enhancing reconstruction quality and providing robust Self-Supervised Learning (SSL) supervision for slide-level pretraining. The MSVQ-based offline tokenizer serves as a stable and effective supervisory signal for SSL pretraining and is compatible with both ABMIL and WSI-Transformer architectures. Extensive experiments across multiple datasets validate that the proposed approach achieves state-of-the-art performance in whole slide image (WSI) analysis tasks.

**Questions:**

Could you please furnish a concise description of Figures 7, 8, 9, and 10?

**Ethical Concerns:**

["NO or VERY MINOR ethics concerns only"]

**Final Justification:**

After reviewing all the comments and responses, I tend to keep my initial score.

**Limitations:**

Yes

**Paper Formatting Concerns:**

No.

**Quality:**

3

**Strengths And Weaknesses:**

Strengths:
-  The paper introduces a VQ-based feature distillation framework to address the fundamental trade-off between efficiency and representational richness in WSI foundation models. It may offer a scalable and effective solution for high-performing pathology FMs
- The methodology is well-detailed, explaining the VQ learning, MSVQ, and the slide-level SSL objectives.
- The paper is generally well-structured.


Weaknesses:
- The novelty of the work appears to be relatively incremental, as both Vector Quantization (VQ) and autoencoders are well-established techniques for information compression.
- The ablation studies section lacks a more thorough and detailed discussion.
- Certain symbols—such as those presented in Algorithm 1—require further specification and clarification.

---

> ### Author Rebuttal · Authors · 2025-07-30
>
> We thank the reviewer for the insightful comments and appreciate the reviewer's concerns.
>
> >W1:The novelty of the work appears to be relatively incremental, as both Vector Quantization (VQ) and autoencoders are well-established techniques for information compression.
>
> While our work builds upon established techniques such as VQ and SSL, our core contribution lies in **unifying two longstanding challenges in WSI foundation model pretraining** through a coherent and scalable **multi-scale VQ framework**:
>
> **(1) Addressing the CLS token bottleneck via patch-level (16x16) VQ:**
>
> CLS token–based representations often discard fine-grained spatial information, leading to suboptimal performance on downstream pathology tasks. To mitigate this, we apply patch-level (patch size = 16) VQ/autoencoding to preserve local semantics while significantly reducing spatial dimensionality. This allows us to compress the token space without resorting to a single global embedding.
>
> **(2) Improving the SPT (Slide-level Pretraining) objective via tile-level (224x224) VQ:**
>
> Traditional masked autoencoding (e.g., MAE) focuses on reconstructing pixel inputs, which is infeasible for WSI-scale images due to GPU memory limits. We instead mask and predict **intermediate tile-level feature representations**, but just like MAE learn too many low-frequency feature of raw-pixel image [1], directly using feature-level targets also not align well with task-specific semantics.
>
> To resolve this, we adopt a **tile-level (tile size = 224) VQ tokenizer** that discretizes feature targets into informative codebook entries. This approach enables more stable and effective slide-level representation learning compared to raw pixel or continuous-feature reconstruction.
>
> **Unified framework via Multi-Scale VQ:**
>
> Rather than treating patch-level compression and slide-level representation learning as disjoint processes, we introduce a **Multi-Scale VQ architecture** that links both levels: patch-level VQ captures local detail, while tile-level VQ supports robust and interpretable global objectives. This unified view of representation compression and learning makes our approach both practically scalable and theoretically grounded for WSI tasks.
>
> In addition, for addressing issue (1)—the CLS token bottleneck—we tackle a fundamental limitation in transformer-based slide-level models: **insufficient input rank**, which constrains their capacity to represent complex WSI data. As noted in [2], the rank of the attention matrix in a standard transformer is theoretically bounded:
>
> $$
> \text{rank}(QK^\top) \leq \min(\text{rank}(Q), \text{rank}(K)) = \min(n, d),
> $$
> where n is the sequence length and d is the feature dimension. When using a single CLS token (or averaged tile-level CLS features), the actual rank is severely limited. Though in [2] authors try to improve the rank via local attention and positional embedding, the input rank limitation still exists. For example, we observe that in **Conch**, the effective rank of WSI embeddings using tile-level CLS is approximately **50**, and in **UNI**, it is around **400**—both significantly lower than the typical WSI sequence length (often exceeding **5,000** tokens).
>
> | Model | Rank (Mean ± Std) | WSI Length (Mean ± Std) | AUC Transformer |
> | --- | --- | --- | --- |
> | **Conch** | 53.6 ± 6.2 | 3356.1 ± 1840.6 | 0.861 |
> | **UNI** | 367.3 ± 109.7 | 3356.1 ± 1840.6 | 0.882 |
> | **Ours** | 765.4 ± 88.3 | 3356.1 ± 1840.6 | 0.910 |
>
> In contrast, our method reconstructs spatial tokens via VQ-based representations, enabling **a much higher-rank representation** of the WSI content. We can also **increase the feature dimensionality (e.g., to 2048)**, which is more effective than simply up-projecting CLS features from 1024 using a linear layer.
> For context, in standard ViT-Large (e.g., ViT-L/16 with 224×224 inputs), a 1024-d embedding is used for just 196 tokens. Large language models (e.g., LLaMA, Qwen ranging from 3~7B) employ embeddings of 2048–4,096 dimensions to represent typical question lengths. By analogy, for long-sequence AI applications such as WSIs, **our approach provides a scalable and rank-preserving alternative perspective**, and may generalize to other domains facing similar sequence-length bottlenecks.
>
> >W2: The ablation studies section lacks a more thorough and detailed discussion.
>
> Sorry for the non-thorough discussion on ablation section due to the page limitation. Here we provide an detailed analysis and will include this in next version:
>
> **Performance vs. Complexity across  reduction methods**:
> Figure 5 illustrates the trade-off between performance and complexity for different token reduction methods, evaluated on the BRACS dataset.
>
> PCA: Reduces token dimensions by projecting them onto a low-rank linear subspace, preserving variance but potentially discarding discriminative details.The F1-score reaches about 0.70.
>
> Average Pooling: Aggregates tokens by computing their mean, leading to compact representations but often oversmoothing critical local information. We have conducted pooling original 196=14x14 token features into 4x4 and 7x7 tokens, resulting in about 10 and 2 times token reduction. However, the result only show similar performance compared to original CLS token.
>
> CLS and overall average-pooling: as performed in virchow, the CLS and average-pooling token can complement a little to each other and resulting in around 1.0 point improvement.
>
> We also performed token selection via CLS token attention top-k ranking. The selected top64 can reach a F1-score around 0.72 but is still too heavy in computation cost. The selected top16 can reach a F1-score of 0.70, can be seen as a trade-off, but lose too much on performance.
>
> Our proposed VQ-based method achieves the best balance, attaining the highest F1-score of 0.73 while significantly reducing token complexity. In contrast, other strategies demonstrate varying degrees of compromise between accuracy and computational cost:
>
> Our ablation shows that VQ not only improves computational efficiency but also maintains or enhances discriminative capacity, compared to conventional reduction techniques.
>
> >W3: Certain symbols—such as those presented in Algorithm 1—require further specification and clarification.
>
> Thank you for pointing this out. We will revise Algorithm 1 to clarify the notation and added detailed definitions for all variables used in next version.
>
> >Q1:  furnish a concise description of Figures 7, 8, 9, and 10
>
> Thanks a lot for this advise and we will furnish it in next version.
>
>
> Ref:
>
> [1] Beit: Bert pre-training of image transformers
>
> [2] Rethinking Transformer for Long Contextual Histopathology Whole Slide Image Analysis

---

### Official Review · Reviewer_LDx8 · 2025-07-04

**Clarity:** 3
**Significance:** 2
**Originality:** 2
**Rating:** 4
**Confidence:** 4

**Summary:**

Processing whole-slide pathology images poses a computational challenge due to their sheer size. When working on slide-level tasks, the slides are divided into thousands of usually 224x224 tiles, each having 196 16x16 patches. Most models summarize whole tiles into a single [CLS] token for slide-level prediction, which is computationally efficient but loses rich patch-level information. As using large tokens from each class is prohibitive, the authors introduce a Vector Quantization (VQ) scheme, where a fixed-size codebook is learned and used to summarize each patch.

Furthermore, two related techniques for slide-level pretraining are introduced and feature alignment is performed when converting the patch-level codes into a final slide-level encoding. The effectiveness of the method is well illustrated on 3 different tasks (tumor classification, mutation prediction, survival prediction) and it performs favorably compared to many competing approaches.

**Questions:**

I've covered my concerns about the novelty of the paper in the "Weaknesses" section.

As of now, I would strongly advocate for the acceptance of this paper in a medically focused conference, but have my doubts as to how interesting it is for the machine learning community at large. I am still veering on the side of acceptance due to the strong aggregation and evaluation which makes this contribution a strong model and baseline for other researchers and practicioners to base their work on. If, after the rebuttal, I am convinced that there is more novelty in the paper than I've noticed so far, I'd gladly raise my score from 4 to 5.

**Ethical Concerns:**

["NO or VERY MINOR ethics concerns only"]

**Final Justification:**

I still find the novelties contained in the approach to be too limited as I stated in my initial reviews, despite the interesting points raised by the authors during rebuttal. The most interesting part of the rebuttal: The rank based perspective, is not present in the paper and was not designed for, and needs a deeper look.

**Limitations:**

The coverage of limitations is token, authors state the main weakness of VQ is that "it needs more training data" and that "there is information loss although it is acceptable". These claims should be further substantiated in supplementary material and the grammar of the conclusion section reviewed.

**Paper Formatting Concerns:**

-

**Quality:**

4

**Strengths And Weaknesses:**

Strengths

* The problem being tackled and the proposed VQ-based solution is explained and illustrated clearly, and solves a tangible problem in WSI-based tasks.
* Related work is covered sufficiently.
* Evaluation on tasks and competing models is extensive.

Weaknesses

* There are no particularly novel technical contributions in the paper. It aggregates several techniques which have been previously studied:
* VQ is a well known technique, as covered in the paper's related work.
* The WSI-Transformer self-supervised learning objective which tries to predict the codebook assignment of tokens missing in a sequence is very similar to masked token prediction, which is fairly well known [1].
* The ABMIL self-supervised learning objective which tries to predict the global distribution of token assignments is similar to [2], where predicting the code distribution of cross-view images is a task.
* The token-alignment during conv-upsampling performed while summarizing patches into tiles is straightforward.

[1] Bao, Hangbo, et al. "Beit: Bert pre-training of image transformers." arXiv preprint arXiv:2106.08254 (2021).
[2] Gidaris, Spyros, et al. "Moca: Self-supervised representation learning by predicting masked online codebook assignments." TMLR (2024).

---

> ### Author Rebuttal · Authors · 2025-07-30
>
> We thank the reviewer for the insightful comments and appreciate the reviewer's concerns regarding the novelty of our method. While our work builds upon established techniques such as VQ and SSL, our core contribution lies in **unifying two longstanding challenges in WSI foundation model pretraining** through a coherent and scalable **multi-scale VQ framework**:
>
> **(1) Addressing the CLS token bottleneck via patch-level (16x16) VQ:**
>
> CLS token–based representations often discard fine-grained spatial information, leading to suboptimal performance on downstream pathology tasks. To mitigate this, we apply patch-level (patch size = 16) VQ/autoencoding to preserve local semantics while significantly reducing spatial dimensionality. This allows us to compress the token space without resorting to a single global embedding.
>
> **(2) Improving the SPT (Slide-level Pretraining) objective via tile-level (224x224) VQ:**
>
> Traditional masked autoencoding (e.g., MAE) focuses on reconstructing pixel inputs, which is infeasible for WSI-scale images due to GPU memory limits. We instead mask and predict **intermediate tile-level feature representations**, but just like MAE learn too many low-frequency feature of raw-pixel image [1], directly using feature-level targets also not align well with task-specific semantics.
>
> To resolve this, we adopt a **tile-level (tile size = 224) VQ tokenizer** that discretizes feature targets into informative codebook entries. This approach enables more stable and effective slide-level representation learning compared to raw pixel or continuous-feature reconstruction.
>
> **Unified framework via Multi-Scale VQ:**
>
> Rather than treating patch-level compression and slide-level representation learning as disjoint processes, we introduce a **Multi-Scale VQ architecture** that links both levels: patch-level VQ captures local detail, while tile-level VQ supports robust and interpretable global objectives. This unified view of representation compression and learning makes our approach both practically scalable and theoretically grounded for WSI tasks.
>
> In addition, for addressing issue (1)—the CLS token bottleneck—we tackle a fundamental limitation in transformer-based slide-level models: **insufficient input rank**, which constrains their capacity to represent complex WSI data. As noted in [2], the rank of the attention matrix in a standard transformer is theoretically bounded:
>
> $$
> \text{rank}(QK^\top) \leq \min(\text{rank}(Q), \text{rank}(K)) = \min(n, d),
> $$
> where n is the sequence length and d is the feature dimension. When using a single CLS token (or averaged tile-level CLS features), the actual rank is severely limited. Though in [2] authors try to improve the rank via local attention and positional embedding, the input rank limitation still exists. For example, we observe that in **Conch**, the effective rank of WSI embeddings using tile-level CLS is approximately **50**, and in **UNI**, it is around **400**—both significantly lower than the typical WSI sequence length (often exceeding **5,000** tokens).
>
> | Model | Rank (Mean ± Std) | WSI Length (Mean ± Std) | AUC Transformer |
> | --- | --- | --- | --- |
> | **Conch** | 53.6 ± 6.2 | 3356.1 ± 1840.6 | 0.861 |
> | **UNI** | 367.3 ± 109.7 | 3356.1 ± 1840.6 | 0.882 |
> | **Ours** | 765.4 ± 88.3 | 3356.1 ± 1840.6 | 0.910 |
>
> In contrast, our method reconstructs spatial tokens via VQ-based representations, enabling **a much higher-rank representation** of the WSI content. We can also **increase the feature dimensionality (e.g., to 2048)**, which is more effective than simply up-projecting CLS features from 1024 using a linear layer.
>
> For context, in standard ViT-Large (e.g., ViT-L/16 with 224×224 inputs), a 1024-d embedding is used for just 196 tokens. Large language models (e.g., LLaMA, Qwen ranging from 3~7B) employ embeddings of 2048–4,096 dimensions to represent typical question lengths. By analogy, for long-sequence AI applications such as WSIs, **our approach provides a scalable and rank-preserving alternative perspective**, and may generalize to other domains facing similar sequence-length bottlenecks.
>
> Ref:
>
> [1] Beit: Bert pre-training of image transformers
>
> [2] Rethinking Transformer for Long Contextual Histopathology Whole Slide Image Analysis

---

> > ### Comment · Reviewer_LDx8 · 2025-08-06
> >
> > I thank the authors for their detailed rebuttal. I appreciate the clarifications provided for points (1) and (2), as well as the utility of the proposed method as an end-to-end framework. The rank-based perspective is indeed interesting and adds value to the paper, though it seems to emerge naturally rather than being an explicit design goal. As such, I still feel that the overall level of novelty is somewhat limited and believe my initial score remains appropriate.

---

> > > ### Author Response · Authors · 2025-08-07
> > > **Thanks**
> > >
> > > Thank you very much for your thoughtful feedback and for acknowledging our clarifications and the utility of our proposed end-to-end framework.
> > > We appreciate your recognition of the rank-based perspective and your constructive comments regarding the novelty of our work.
> > > Your insights are valuable to us and will help guide our future research.

---

### Decision · Program_Chairs · 2025-09-17

**Decision:**

Accept (poster)

**Comment:**

This submission tackles WSI data processing with an aim to address the trade-off between computational efficiency and information richness of representations. The developed foundation model has been applied and validated on downstream tasks of tumor classification, mutation prediction, and survival prediction. All the reviewers acknowledged the importance of the paper's research topic, and appreciate the workload of experiments. Initial weaknesses pointed out by the reviewers included inadequacy of technical contribution (e.g., VQ and autoencoders are existing), unclear motivation/justification of the method design (e,g., reason of using SSL and VQ), and details in writing (e.g., reporting training cost). The authors' rebuttal has successfully addressed the raised comments and sufficiently alleviated the major concerns. As a result, all reviewers positively rated Borderline Accept. The authors may consider to add the clarifications provided in the rebuttal into the final version.